# The Statistical Complexity of Early-Stopped Mirror Descent

**Tomas Vaškevičius[1], Varun Kanade[2], Patrick Rebeschini[1]**
[1] Department of Statistics, [2] Department of Computer Science
University of Oxford
{tomas.vaskevicius, patrick.rebeschini}@stats.ox.ac.uk
varunk@cs.ox.ac.uk

## Abstract

Recently there has been a surge of interest in understanding implicit regularization properties of iterative gradient-based optimization algorithms. In this paper, we study the statistical guarantees on the excess risk achieved by early-stopped unconstrained mirror descent algorithms applied to the unregularized empirical risk with the squared loss for linear models and kernel methods. By completing an inequality that characterizes convexity for the squared loss, we identify an intrinsic link between offset Rademacher complexities and potential-based convergence analysis of mirror descent methods. Our observation immediately yields excess risk guarantees for the path traced by the iterates of mirror descent in terms of offset complexities of certain function classes depending only on the choice of the mirror map, initialization point, step-size, and the number of iterations. We apply our theory to recover, in a clean and elegant manner via rather short proofs, some of the recent results in the implicit regularization literature, while also showing how to improve upon them in some settings.[1]

## 1 Introduction

In a typical statistical learning setup, we observe a dataset $D_n$ of $n$ input-output pairs $(x_i, y_i) \in \mathbb{R}^d \times \mathbb{R}$ sampled i.i.d. from some unknown distribution $P$. When learning with respect to the quadratic loss, the goal is to output a function $\widehat{g} = \widehat{g}(D_n) : \mathbb{R}^d \to \mathbb{R}$ which minimizes the *risk* $R(\widehat{g})$ defined as follows, for any square-integrable function $g$:

$$R(g) = \mathbb{E}_{(X,Y) \sim P} \left[ (g(X) - Y)^2 \right].$$

Among the most studied statistical estimators is the *empirical risk minimization* (ERM) algorithm, which given a function class $\mathcal{G}$ outputs a function $\widehat{g}_{\mathcal{G}} = \widehat{g}_{\mathcal{G}}(D_n)$ defined as

$$\widehat{g}_{\mathcal{G}} \in \underset{g \in \mathcal{G}}{\arg\min}\, R_n(g), \quad \text{where} \quad R_n(g) := \frac{1}{n} \sum_{i=1}^{n} (g(x_i) - y_i)^2, \tag{1}$$

in some cases with a regularization penalty term added to the optimization objective $R_n(g)$, such as $\ell_p$ norm of the model parameters. We consider the *non-realizable* or *agnostic* setting, i.e., the case in which there is no assumption that $\mathbb{E}[Y|X]$ is determined by a well-specified model from a reference class of functions. In the agnostic case, a key performance measure of an estimator $\widehat{g}$ is its *excess risk* with respect to some reference class of functions $\mathcal{F}$:

$$\mathcal{E}(\widehat{g}, \mathcal{F}) = R(\widehat{g}) - \inf_{f \in \mathcal{F}} R(f).$$

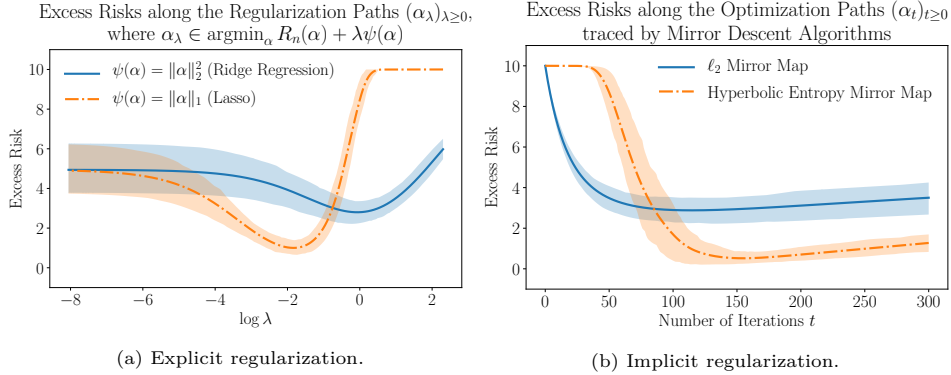

(a) Explicit regularization.             (b) Implicit regularization.

Figure 1: Consider a distribution $P$ such that $X \sim N(0, I_d)$ and $Y|X = x \sim \langle \alpha', x \rangle + N(0, 5^2)$ for some parameter $\alpha' \in \mathbb{R}^d$. Fix $n = 200, d = 100$ and let $\alpha'$ be a 10-sparse vector with non-zero entries equal to $\pm 1$. Due to the sparsity of $\alpha'$, explicit regularization via $\ell_1$ penalization results in a class of models $(\alpha_\lambda)_{\lambda \geq 0}$ that at its minimum achieves significantly lower risk than the class of models generated via $\ell_2$ penalization (cf. Figure 1a). Figure 1b demonstrates a similar phenomenon from an implicit regularization point of view. Due to the sparsity of $\alpha'$, the choice of a hyperbolic entropy mirror map (cf. Section 4) yields an optimization path that at its minimum achieves excess risk nearly an order of magnitude lower than the path generated by the vanilla gradient descent updates. In the plot above, the solid lines denote means over 100 runs whereas the shaded regions correspond to the $10^{th}$ and the $90^{th}$ percentiles.

Traditionally, in learning theory, statistical and computational properties of ERM estimators have been considered separately. From a statistical point of view, localized complexity measures have become a default tool in statistical learning theory and empirical processes theory for controlling the excess risk of ERM algorithms $\widehat{g}_\mathcal{G}$ with respect to the function class $\mathcal{G}$ itself, i.e., for controlling $\mathcal{E}(\widehat{g}_\mathcal{G}, \mathcal{G})$ [7, 20]. A rich and general theory regarding these complexity measures has been developed and used to provide excess risk bounds in both classification and regression settings, yielding minimax-optimal results in several cases. Such complexity measures depend on combinatorial or geometric parameters of interest, such as the VC-dimension or eigenvalue decay of the kernel matrix and, in particular, they serve as a guiding principle to choose a suitable *explicit regularizer* for a set of candidate models $(\widehat{g}_{\mathcal{G}_\lambda})_{\lambda \in \Lambda}$, where $\lambda \in \Lambda$ is a hyper-parameter that controls the amount of regularization. In practice, some $\lambda^\star \in \Lambda$ is then chosen via some model selection procedure such as cross-validation, aiming to select a model with the smallest risk. From a computational point of view, computing the estimators $(\widehat{g}_{\mathcal{G}_\lambda})_{\lambda \in \Lambda}$ can be done by solving the corresponding optimization problems defined in Equation (1), one for each $\lambda \in \Lambda$. An appealing aspect of this approach is that the design and analysis of efficient optimization algorithms, exploiting the geometry of $G_\lambda$ that arises from the the structure of the model as well as the distribution $P$, can be done independently of the statistical analysis of its performance.

Recent years have also witnessed an increased interest in directly studying the statistical properties of models trained by gradient-based methods, particularly in relation to the notions of *implicit regularization* and *early stopping*. For a family of functions $\mathcal{G} = \{g_\alpha : \alpha \in \mathbb{R}^m\}$ parametrized by a vector $\alpha$, such methods are fully characterized by the initialization point $\alpha_0$ and an update rule, which given $\alpha_t$ and the gradient of the empirical risk at $\alpha_t$, generates the next iterate $\alpha_{t+1}$, yielding a set of candidate estimators $(\widehat{g}_{\alpha_t})_{t \geq 0}$. Early stopping has an effect akin to *explicit* regularization discussed above, and the *stopping time* $t^\star$ can be chosen in practice via cross-validation, just as in the case of choosing the explicit regularization parameter $\lambda^\star$ corresponding to the best model among $(\widehat{g}_{\mathcal{G}_\lambda})_{\lambda \in \Lambda}$. In modern large-scale machine learning applications, early stopping is often the preferred way to perform model selection, since obtaining a new model is as cheap as performing a step of gradient descent, as opposed to solving a new optimization problem with a different regularization parameter. In Figure 1, we demonstrate that different choices of optimization algorithms applied to the unregularized empirical risk $R_n$ yield different statistical performance along the optimization

path $(\widehat{g}_{\alpha_t})_{t\geq 0}$, in a similar way that a choice of an explicit regularizer affect the statistical performance along the corresponding regularization path.

It is by now well understood that changing the update rule that generates the sequence $(\widehat{g}_{\alpha_t})_{t\geq 0}$, e.g., by changing the optimization algorithm or parametrization of the model class, can directly affect both the statistical properties of the iterates $\widehat{g}_{\alpha_t}$, as well as computational properties, such as an upper-bound on the optimal stopping time $t^\star$. However, most of the literature has focused on the investigation of vanilla gradient descent updates: $\alpha_{t+1} = \alpha_t - \eta \nabla_{\alpha_t} R_n(\widehat{g}_{\alpha_t})$ (cf. Section 2.1). The existing theory does not easily generalize to other update rules corresponding to different problem geometries. A general theory that connects the notion of early stopping for a more general class of update rules with the well-established theory of localized complexities is still missing. More broadly, a general "language" to reason about the statistical properties of trajectories traced by optimization algorithms applied to the unregularized empirical risk is still lacking.

In this paper, we study a *family* of update rules given by the mirror descent algorithm [27, 9]. Mirror descent, which includes vanilla gradient descent as a special case, is increasingly becoming the tool of choice in optimization and machine learning, applied well beyond the traditional settings of convex optimization and online learning. Among the properties that make mirror descent appealing are its ability to exploit non-Euclidean geometries via properly designed mirror maps, the fact that the algorithm admits a general potential-based convergence analysis in terms of Bregman divergences, and its ability to represent a large class of algorithms in a unified and well-developed framework.

We consider a setting where conditionally on the observed data $D_n$ there exists a matrix $Z \in \mathbb{R}^{n\times m}$ such that the parametric family of functions $\{g_\alpha : \alpha \in \mathbb{R}^m\}$ satisfies $g_\alpha(x_i) = (Z\alpha)_i$ for all $i = 1, \ldots, n$. As special cases, our setup admits linear regression and kernel methods (cf. Section 4), cornerstones of modern statistics and machine learning. Our work reveals an inherent connection between the statistical properties of the mirror descent iterates $(\widehat{g}_{\alpha_t})_{t\geq 0}$ and the notion of offset Rademacher complexity [23]. Consequently, our work unearths a simple and elegant way to simultaneously analyze upper-bounds on the stopping time $t^\star$, as well as the excess risk $\mathcal{E}(\widehat{g}_{\alpha_t}, \mathcal{F})$ for all $t \leq t^\star$ in terms of the mirror map, the initialization point $\alpha_0$, the step-size, and the function class $\mathcal{F}$. Through a simple one page analysis, we are able to rederive (nearly identical) results from prior work connecting early stopping and (optimal) statistical performance that previously involved several pages of low-level arguments.[2] Additionally, in the well-studied case of Euclidean gradient descent, our work improves upon the prior results connecting early stopping to localized complexity measures [32, 42] by providing upper-bounds on the expected excess risk without any distributional assumptions on $P$ other than boundedness (cf. Section 2.1).

## 1.1 Background

In this section, we describe offset Rademacher complexities, a form of localization based on mathematical machinery that is more suitable for our setting than that used to develop classical localized complexities [7, 20] (see [39] for an extended discussion). Then, we define the mirror descent updates and outline a well-known potential-based proof of its convergence

In what follows, we let $\|g-f\|_n^2 = \frac{1}{n}\sum_{i=1}^n (g(x_i)-f(x_i))^2$ and $\|g-f\|_P^2 = \mathbb{E}[(g(X)-f(X))^2]$ denote the empirical and population $\ell_2$ distances between functions $g$ and $f$, respectively. Further, given a function class $\mathcal{F}$, we denote by $g_{\mathcal{F}} \in \mathcal{F}$ a function that attains risk equal to $\inf_{g\in\mathcal{F}} R(g)$.[3] A table of notation is provided in the full version of this paper [39].

**Offset Rademacher complexities.** When learning with the quadratic loss, a theory of localization based on shifted Rademacher processes was developed by Liang et al. [23] inspired by prior work in online learning [30]. The use of shifted empirical processes in order to bypass technicalities present in the classical localization arguments date back at least to [41] and have recently found applications in cross-validation [21], classification [47] and PAC-Bayes bounds [44]. For a function class $\mathcal{G}$, a dataset $D_n$, an independent sequence of Rademacher random variables $\sigma_1, \ldots, \sigma_n$, and any $c \geq 0$, the *empirical offset Rademacher complexity* is defined as, conditionally on the observed data $D_n$:

$$\mathfrak{R}_n(\mathcal{G}, c) = \mathbb{E}_{\sigma_1, \ldots, \sigma_n}\left[ \sup_{g \in \mathcal{G}} \left\{ \frac{1}{n} \sum_{i=1}^{n} (2\sigma_i g(x_i) - cg(x_i)^2) \right\} \right].$$

Note that since the terms $-cg(x_i)^2$ are always non-positive, the above notion of complexity is never larger than global Rademacher complexity of the class $\mathcal{G}$, which is recovered with $c = 0$. On the other hand, for any $c > 0$, the quadratic term in the above definition has a localization effect by compensating for the fluctuations in the term involving Rademacher variables (see Section 5.2 and the discussion following Theorem 3 in [23]). Importantly, the theory of localization via offset complexities replaces the Bernstein condition used in the classical theory of localization by the *offset condition* defined below.

**Definition 1** (Offset condition)**.** *A triple $(P, \mathcal{F}, \widehat{g})$ satisfies the offset condition with parameters $\varepsilon \geq 0, c > 0$, if for $D_n \sim P^n$, with probabilty 1, we have $R_n(\widehat{g}) - R_n(g_{\mathcal{F}}) + c\|\widehat{g} - g_{\mathcal{F}}\|_n^2 \leq \varepsilon$.*

The above condition with $\varepsilon = 0$ was introduced in [23] where it was called the *geometric inequality* and shown to hold for ERM estimators over convex classes $\mathcal{F}$ as well as the two-step star estimator [4] over general classes for finite aggregation.[4] A key advantage offered by the theory of offset complexities is that the range of $\widehat{g}$ need not be a subset of $\mathcal{F}$, as long as the offset condition is satisfied. This allows us to consider very general estimators $\widehat{g}$, possibly with non-convex ranges $\mathcal{G}$. In this respect, our work can be seen as showing that early-stopped mirror descent satisfies the offset condition defined above. Once an estimator is shown to satisfy the offset condition, its excess risk $\mathcal{E}(\widehat{g}, \mathcal{F})$ can be controlled in terms of the offset complexity $\mathfrak{R}_n(\mathcal{G} - g_{\mathcal{F}}, c)$. The theory developed in [23] establishes high-probability bounds under the lower-isometry assumption, which can hold even for possibly heavy-tailed classes ([23, Theorem 4]), as well as bounds in expectation under no assumptions other than boundedness ([23, Theorem 3]). The result in expectation states that given $\sup_{g \in \mathcal{G} \cup g_{\mathcal{F}}} |g|_\infty \leq B$ and $\|Y\|_{L_\infty(P)} \leq M$ for some $B, M > 0$, we have

$$\mathbb{E}[\mathcal{E}(\widehat{g}, \mathcal{F})] \leq c_1 \mathbb{E}[\mathfrak{R}_n(\mathcal{G} - g_{\mathcal{F}}, c_2)] + \varepsilon,$$

where $c_1 = (4 + c/2)B + 2M$ and $c_2 = c/(4(B + M)(2 + c))$ and the expectation is taken over datasets $D_n$; here $c$ and $\varepsilon$ are those that appear in the offset condition. The generality of the above result allows us to improve upon the existing bounds in the early stopping literature even for gradient descent updates (cf. Section 2.1).

**Mirror descent.** The key object characterizing the geometry of the mirror descent algorithm is the *mirror map* $\psi$, a strictly convex and differentiable function mapping some open set $\mathcal{D} \subseteq \mathbb{R}^m$ to $\mathbb{R}$ whose gradient is surjective, i.e. $\{\nabla \psi(\alpha) \mid \alpha \in \mathcal{D}\} = \mathbb{R}^m$. By slightly abusing notation, we use $R_n(\alpha) := R_n(g_\alpha)$ to denote the empirical risk of $g_\alpha$. When optimizing the empirical risk $R_n(\alpha)$, the mirror descent updates in continuous and discrete time are given respectively by

$$\frac{d}{dt}\alpha_t = -\left(\nabla^2 \psi(\alpha_t)\right)^{-1} \nabla R_n(\alpha_t) \quad \text{and} \quad \nabla \psi(\alpha_{t+1}) = \nabla \psi(\alpha_t) - \eta \nabla R_n(\alpha_t), \qquad (2)$$

where $\eta > 0$ is the step-size. We remark that the choice $\psi(\alpha) = \frac{1}{2}\|\alpha\|_2^2$ reduces the above updates to gradient descent. A key notion in the analysis of mirror descent algorithms is the *Bregman divergence*, defined as $D_\psi(\alpha', \alpha) = \psi(\alpha') - \psi(\alpha) - \langle \nabla \psi(\alpha), \alpha' - \alpha \rangle$ for all $\alpha', \alpha$ in the domain of $\psi$. By convexity of $\psi$, the Bregman divergence $D_\psi$ is non-negative and enters the analysis of mirror descent algorithms through the following elementary equality:

$$-\frac{d}{dt}D_\psi(\alpha', \alpha_t) = \langle -\nabla R_n(\alpha_t), \alpha' - \alpha_t \rangle. \qquad (3)$$

Let $\bar{\alpha}_t = \frac{1}{t}\int_0^t \alpha_t dt$. In the optimization literature, the above equation can be used to establish that $R_n(\bar{\alpha}_t)$ can get arbitrarily close to $R_n(\alpha')$ from above, for any reference point $\alpha'$. In particular, by convexity of $R_n$, we have $\langle -\nabla R_n(\alpha_t), \alpha' - \alpha_t \rangle \geq R_n(\alpha_t) - R_n(\alpha')$ and so

$$\frac{1}{t}D_\psi(\alpha', \alpha_0) \geq \frac{1}{t}\int_0^t -\frac{d}{ds}D_\psi(\alpha', \alpha_s)ds \geq \frac{1}{t}\int_0^t R_n(\alpha_s) - R_n(\alpha')ds \geq R_n(\bar{\alpha}_t) - R_n(\alpha'), \quad (4)$$

where the last line follows by convexity of $R_n$. Remarkably, the above proof works independently of the choice of the mirror map $\psi$, establishing convergence for a *family* of algorithms in a unified framework. For more information we refer the interested reader to the surveys by Bubeck [10] and Bansal and Gupta [6]. The latter survey focuses entirely on such potential-based proofs in a variety of settings, including acceleration.

## 2 Summary of Techniques and Main Results

We develop a general theory for learning linear models (including kernel machines) with the squared loss that shows how the optimization trajectory of *unconstrained* mirror descent applied to minimize the unregularized empirical risk is *inherently* connected to excess risk guarantees via offset Rademacher complexity. Unlike in most prior work on early stopping, the notion of statistical complexity appears naturally from intrinsic properties of mirror descent applied to the unregularized empirical risk, without invoking lower-level arguments related to concentration to the *fictitious* population version of the algorithm. Furthermore, our theory leads to an explicit characterization of stopping times from the point of view of both optimization and statistics, which directly yields excess risk bounds and allows us to re-derive previously established results, and some new results, in a much simpler fashion.

As discussed in Section 1.1, early-stopped unconstrained iterative algorithms do not easily fit within the mathematical framework of classical localization techniques, partially explaining the scarcity of results connecting localized complexity measures with such algorithms. Offset Rademacher complexities, on the other hand, open up another avenue for establishing such connections via the design of update rules tailored to satisfy the offset condition (cf. Definition 1). Instead of optimizing the empirical risk $R_n$, a natural approach to consider is an application of some iterative optimization algorithm to directly minimize the term appearing in the definition of the offset condition: $\tilde{R}_n^{\alpha', c}(\alpha) = R_n(\alpha) - R_n(\alpha') + c\|g_\alpha - g_{\alpha'}\|_n^2$. For any $c > 0$, the gradient $\nabla_\alpha \tilde{R}_n^{\alpha', c}(\alpha)$ depends on the unknown reference point $\alpha'$ and hence cannot be computed in practice. Remarkably, we show that the mirror descent updates applied to the empirical loss $R_n$ simultaneously *implicitly* minimizes $\tilde{R}_n^{\alpha', 1}$ for *all* reference points $\alpha'$ up to a certain stopping time (which depends on $\alpha'$), while also *staying inside a certain Bregman "ball"* centered at $\alpha'$ up to the corresponding stopping time. While mirror descent was developed within the framework of convex optimization, it has also found applications in a wide range of problems including bandits [1], online learning [19], the k-server problem [11] and metrical task systems [12]. In this respect, our work can be seen as an exposition of yet another example where mirror descent naturally solves a problem outside of its originally intended scope.

The key insight behind our main result is the following identity, linking the potential-based analysis of mirror descent (cf. Section 1.1) to the statistical guarantees derived from offset complexities via the offset condition (cf. Definition 1).

**Lemma 1.** *For any $\alpha, \alpha' \in \mathbb{R}^m$, the following holds:*

$$\langle -\nabla R_n(\alpha), \alpha' - \alpha \rangle = R_n(\alpha) - R_n(\alpha') + \|g_\alpha - g_{\alpha'}\|_n^2.$$

*Proof.* Recall that there exists some $Z \in \mathbb{R}^{n \times m}$ such that for any parameter $\alpha$ we have $g_\alpha(x_i) = (Z\alpha)_i$ (cf. Section 1). Hence, we can write $R_n(\alpha) = \frac{1}{n}\|Z\alpha - y\|_2^2$, where $y \in \mathbb{R}^n$ is a vector with the $i^{th}$ entry equal to $y_i$ and also, $\|g_\alpha - g_{\alpha'}\|_n^2 = \frac{1}{n}\|Z\alpha - Z\alpha'\|_2^2$. The result follows via an application of the equality $2\langle a, b \rangle = \|a\|_2^2 + \|b\|_2^2 - \|a - b\|_2^2$, which holds for any vectors $a, b \in \mathbb{R}^m$. See [39] for full details. $\square$

To appreciate the significance of the above lemma we revisit the potential-based proof of mirror descent presented in Equation (4) in Section 1.1. This time, instead of using the

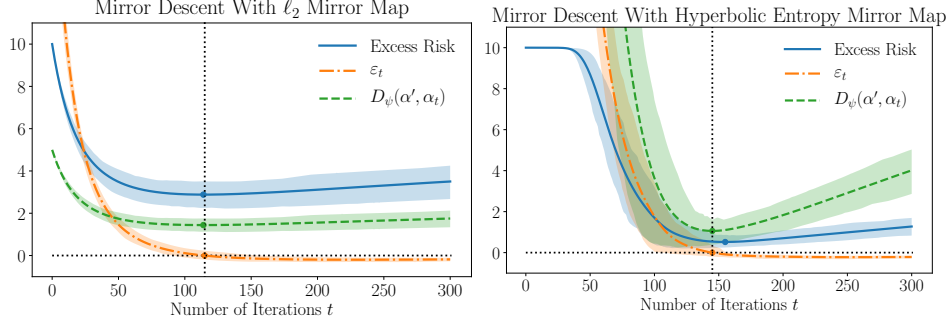

Figure 2: Consider the setting of Figure 1 and let $\varepsilon_t = R_n(\alpha_t) - R_n(\alpha') + \|g_{\alpha_t} - g_{\alpha'}\|_n^2$. The above plots illustrate the following two points. First, there exists a stopping time time $t^\star$ such that $\varepsilon_{t^\star} \approx 0$ (denoted by the vertical dotted line). Hence, the triple $(P, \{g_{\alpha'}\}, g_{\alpha_{t^\star}})$ satisfies the offset condition (cf. Definition 1) with parameters $(c = 1, \varepsilon \approx 0)$. Second, while $\varepsilon_t \geq 0$, the Bregman divergence $D_\psi(\alpha', \alpha_t)$ denoted by the green line is non-increasing. It follows that the estimator $g_{\alpha_{t^\star}}$ is constrained to lie in the set $\{g_\alpha : D_\psi(\alpha', \alpha) \leq D_\psi(\alpha', \alpha_0)\}$, the offset complexity of which can be used to upper-bound the excess risk of interest. Crucially, this type of analysis does not directly rely on the particular form taken by the mirror descent update rules, which bypasses the limitations present in prior work (cf. Section 2.1) and allows us to provide excess risk guarantees for a family of mirror descent algorithms. In the plot above, the solid lines denote means over 100 runs, the dots denote the minimum of each solid line, whereas the shaded regions correspond to the $10^{th}$ and the $90^{th}$ percentiles.

convexity of $R_n$ which gives $\langle -\nabla R_n(\alpha_t), \alpha' - \alpha_t \rangle \geq R_n(\alpha_t) - R_n(\alpha')$, we directly plug in the identity given in Lemma 1 into Equation (3) which yields the following *equality*:

$$-\frac{d}{dt} D_\psi(\alpha', \alpha_t) = R_n(\alpha_t) - R_n(\alpha') + \|g_{\alpha_t} - g_{\alpha'}\|_n^2.$$

The above equation shows that while $R_n(\alpha_t) - R_n(\alpha') + \|g_{\alpha_t} - g_{\alpha'}\|_n^2 > 0$, the iterates of mirror descent stay withing the Bregman ball $\{\alpha \in \mathbb{R}^m : D_\psi(\alpha', \alpha) \leq D_\psi(\alpha', \alpha_0)\}$. At the same time, the integration argument used in Equation (4) establishes that the term $R_n(\alpha_t) - R_n(\alpha') + \|g_{\alpha_t} - g_{\alpha'}\|_n^2$ eventually gets arbitrarily close to 0, and thus the early-stopped mirror descent iterates satisfy the offset condition (cf. Definition 1). For a visual demonstration of the above proof sketch see Figure 2. We provide full details of this argument in the proof of Theorem 1 as well as a discrete-time version in Theorem 2.

**Summary of contributions:**

1. Our work extends the scope of offset Rademacher complexities to a family of early-stopped mirror descent methods. Additionally, we extend the scope of mirror descent to be used as a computationally efficient statistical device in an i.i.d. batch statistical learning setting.

2. Our main results, in a short and transparent way, yield bounds on the excess risk of the iterates of (both continuous-time and discrete-time) mirror descent using offset Rademacher complexities. In contrast to prior work, our arguments require no direct use of low-level mathematical techniques such as symmetrization, peeling, or concentration to the population version of the algorithm.

3. In Section 4, we demonstrate some selected applications of our main results and comment on the connections to the related work therein.

## 2.1 Comparison with Related Work

Statistical and computational properties of unconstrained gradient descent updates have been a subject of intense study over the past two decades, with most of the existing results focusing on the quadratic loss in reproducing kernel hilbert spaces (RKHS) [13, 45, 8, 32, 42]. In contrast to our work, the above work focuses either on bounds in $\|\cdot\|_n^2$ or in $\|\cdot\|_P^2$ norms, which can be arbitrarily smaller than the excess risk considered in our work (see [37, Section

1] for an example). In addition, the analysis in [13, 45, 8, 32] is closely tied to the $\ell_2$ geometry of the gradient descent updates, which allows one to view the algorithm as a particularly simple linear operator acting on the observed labels. Spectral properties of these linear operators are then analyzed as a function of the number of iterations, which can be solved for a stopping time via some form of bias-variance decomposition. Our work, in contrast, enables simultaneously studying a *family* of update rules, characterized by different choices of the mirror map, in a unified framework without relying on their particular form.

One of the primary contributions of our work is the connection between mirror descent iterates and localized complexity measures. To the best of our knowledge, there are only two prior works making connections of a similar nature, albeit only in the setting of Euclidean gradient descent updates, that is, with the choice of the mirror map $\psi(\alpha) = \|\alpha\|_2^2/2$ [32, 42]. Such connections are observed in an algebraic fashion in the former work, while localized complexities appear more naturally in [42], via the analysis of the range of estimators defined by gradient descent iterates up the stopping time. In this respect, the work in [42] is the closest to ours. In Theorem 3, we show how a straightforward application of our main results immediately recovers results similar to the ones obtained in [32, 42].

Beyond the Euclidean setup, interest in understanding the generalization properties of neural networks has sparked research into *implicit* regularization properties of various factorized models. In the context of neural networks, the authors of [17, 22, 3, 43, 16] show that iterates of gradient descent applied to factorized matrix models are implicitly biased towards some sparsity-inducing structure such as low-rankness or low nuclear norm. Such results, however, hold under certain limit statements, such as vanishing initialization or step-size, the number of iterations going to infinity, or no noise in the problem. In the setting of linear regression, matrix factorization models reduce to vector Hadamard product factorizations, where early-stopped gradient descent was shown to yield minimax optimal rates for sparse recovery with the analysis vitally relying on the restricted isometry property [46, 40]. In Theorem 4, we demonstrate a simple analysis of such updates within our framework *without any assumptions* on the design matrix other than bounded columns, yielding an (up to a log factor) minimax optimal algorithm for in-sample linear prediction under $\ell_1$ norm constraints.

Implicit regularization properties of mirror descent have recently attracted a considerable amount of attention; however, most results in this area either focus on optimization guarantees that do not provide any direct link to statistical guarantees on out-of-sample prediction [18, 5], or establish a connection to statistics via some forms of explicit regularization [38]. The work [38] shows connections between the iterates on the entire path and the solutions on the regularization path for a suitable regularized risk minimization problem. In Theorem 5, we show how the analysis of such problems naturally fit within our framework. Yet other papers have used early stopping to solvers applied directly to appropriately constrained problems and regularization-promoting structures encoded directly into the loss function [26].

Recent work has also focused on providing statistical guarantees for iterates generated via gradient descent updates in stochastic [35, 25, 28, 2], accelerated [14, 29], and distributed settings [24, 33, 34]. These works provide statistical guarantees without establishing connections to localized complexity measures; we anticipate such connections to be studied within our framework in future work, for a family of mirror descent algorithms.

## 3 Main Results

We first state and prove a continuous-time version of our main theorem, which demonstrates the key ideas behind our approach in the simplest setting. The first part of the theorem shows that the iterates of mirror descent stay within a certain Bregman ball up to the prescribed stopping time $t^\star$. The second part of the theorem immediately establishes that when the parametrization given by $\alpha \in \mathbb{R}^m$ is independent of the data, the early-stopped estimator $g_{\alpha_{t^\star}}$ satisfies the offset condition (cf. Definition 1) with parameters $c = 1$ and any $\varepsilon > 0$.[5] For the applications we consider, we choose $\varepsilon$ to match the complexity measure of interest and

recover the statistical-computational trade-offs consistent with the previous results in the literature. In particular, $t^\star = O(D_\psi(\alpha', \alpha_0)/\varepsilon)$, so that achieving higher statistical accuracy requires more computational power. Finally, we note that the dependence of $t^\star$ on the unknown radius $D_\psi(\alpha', \alpha_0)$ is unavoidable purely from an optimization point of view.

**Theorem 1.** *Consider the continuous-time mirror descent updates given in Equation* (2). *Let $\alpha_0$ be the initialization point, $\alpha'$ be any chosen reference point, and fix any $\varepsilon > 0$. Then, there exists a stopping time $t^\star = t^\star(D_n, \psi, \alpha_0, \alpha') \le 2D_\psi(\alpha', \alpha_0)/\varepsilon$ such that:*

    *1. For all $0 \le t \le t^\star$, $g_{\alpha_t} \in \mathcal{G}(\psi, \alpha_0, \alpha') = \{g_\alpha \in \mathbb{R}^m : D_\psi(\alpha', \alpha) \le D_\psi(\alpha', \alpha_0)\}$.*

    *2. At the stopping time $t^\star$, we have $R_n(\alpha_{t^\star}) - R_n(\alpha') + \|g_{\alpha_{t^\star}} - g_{\alpha'}\|_n^2 \le \varepsilon$.*

*Proof.* To simplify the notation let $\delta_t = R_n(\alpha_t) - R_n(\alpha')$ and $r_t = \|g_{\alpha_t} - g_{\alpha'}\|_n^2$. Combining Equation (3) with Lemma 1 we have $-\frac{d}{dt}D_\psi(\alpha', \alpha_t) = r_t + \delta_t$. Let $T = 2D_\psi(\alpha', \alpha_0)/\varepsilon$. Integrating both sides of the above equality we obtain

$$D_\psi(\alpha', \alpha_0) - D_\psi(\alpha', \alpha_T) = \int_0^T -\frac{d}{dt}D_\psi(\alpha', \alpha_t)dt = \int_0^T (r_t + \delta_t)dt$$

$$\implies \inf_{0 \le t \le T}\{r_t + \delta_t\} \le \frac{1}{T}\int_0^T (r_t + \delta_t)dt \le \frac{D_\psi(\alpha', \alpha_0)}{T} \le \frac{\varepsilon}{2}.$$

It follows that the following infimum is well defined: $t^\star = \inf\{0 \le t \le T \mid r_t + \delta_t \le \varepsilon\}$. Hence, $r_{t^\star} + \delta_{t^\star} \le \varepsilon$ and for all $0 \le t \le t^\star$ we have

$$D_\psi(\alpha', \alpha_0) - D_\psi(\alpha', \alpha_t) = \int_0^t (r_t + \delta_t)dt \ge t\varepsilon \ge 0.$$

The above inequality implies that $D_\psi(\alpha', \alpha_t) \le D_\psi(\alpha', \alpha_0)$, which concludes our proof. $\quad\square$

In the next theorem, we prove an equivalent result in discrete-time. Let $\|\cdot\|$ denote any norm. We say that $R_n$ is $\beta$-smooth with respect to $\|\cdot\|$ if $R_n(\alpha') \le R_n(\alpha) + \langle \nabla R_n(\alpha), \alpha' - \alpha \rangle + \frac{\beta}{2}\|\alpha - \alpha'\|^2$ for any $\alpha, \alpha'$ in the domain of $R_n$. We also say that the mirror map $\psi$ is $\rho$-strongly convex with respect to $\|\cdot\|$ if for any $\alpha, \alpha'$ we have $D_\psi(\alpha', \alpha) \ge \frac{\rho}{2}\|\alpha' - \alpha\|^2$. The proof of the below theorem is presented in the extended version of this paper [40].

**Theorem 2.** *Consider the discrete-time mirror descent updates given in Equation* (2). *Suppose that $R_n$ is $\beta$-smooth and $\psi$ is $\rho$-strongly convex with respect to some norm $\|\cdot\|$. Let $\alpha_0$ be the initialization point, $\alpha'$ be any reference point, $\eta \le \rho/\beta$, and fix any $\varepsilon > 0$. Then, there exists a stopping time $t^\star = t^\star(D_n, \psi, \alpha_0, \alpha', \eta) \le (D_\psi(\alpha', \alpha_0) + \eta R_n(\alpha'))/(\eta\varepsilon)$ such that:*

    *1. For all $0 \le t \le t^\star$, $g_{\alpha_t} \in \mathcal{G}(\psi, \alpha_0, \alpha', \eta) = \{g_\alpha : D_\psi(\alpha', \alpha) \le D_\psi(\alpha', \alpha_0) + \eta R_n(\alpha')\}$.*

    *2. At the stopping time $t^\star$, we have $R_n(\alpha_{t^\star}) - R_n(\alpha') + \|g_{\alpha_{t^\star}} - g_{\alpha'}\|_n^2 \le \varepsilon$.*

We now briefly comment on the above theorem. First, the step size condition $\eta \le \rho/\beta$ and the number of iterations $O(1/\varepsilon)$ needed to reach a desired level of accuracy are identical to the guarantees proved in purely convex optimization settings (cf. Theorem 4.4 in [10]). On the other hand, comparing Theorems 1 and 2, in the discrete setting we pay a price of $\eta R_n(\alpha')$ in the radius of the Bregman ball where our early-stopped estimator lies. This is consistent with prior work in the early stopping literature, where such an expansion of the radius dependent on the noise level[6] propagates into the resulting bounds (cf. definition of $C$ in Theorem 1 in [42]). Our work, on the other hand, allows for a more fine-grained control of statistical-computational trade-offs via a selection of a small enough step-size $\eta$.

## 4   Selected Applications of the Main Results

In this section, we discuss three selected applications of our main theorems. See the full version of this paper for proofs, comparisons with related work and additional context [39].

**Early stopping for non-parametric regression.** Let $P$ be *any* distribution supported on $\mathcal{X} \times [-M, M]$ and let $k : \mathcal{X} \times \mathcal{X} \to [0, \infty)$ be a Mercer kernel which induces a Hilbert space of functions $\mathcal{H}$ equipped with norm $\|\cdot\|_{\mathcal{H}}$. Assume that $\sup_{x \in \mathcal{X}} k(x, x) \leq L$ for some constant $L > 0$ and, conditionally on the observed data, denote by $K \in \mathbb{R}^{n \times n}$ a matrix such that $K_{ij} = k(x_i, x_j)$. Such a setup is standard in the literature and we refer the interested reader to the book by Scholkopf and Smola [36] for more background on RKHS.

In the theorem below, we consider mirror descent updates with $\psi(\alpha) = \alpha^\mathsf{T} K \alpha$ (cf. [39] for justification when $K$ is singular) with $\alpha_0 = 0$ and $\eta \leq 1 \wedge 1/\lambda_{max}(K/n)$, where $\lambda_{\max}(K/n)$ is the maximum eigenvalue of $K/n$. In particular, $Z = K$ and $R_n(\alpha) = \|Z\alpha - y\|_2^2/n$.

**Theorem 3.** *Consider the setup described above. Fix any $R > 0$ and let $\mathcal{F}_R = \{h \in \mathcal{H} : \|h\|_{\mathcal{H}} \leq R\}$. There exists a data-dependent stopping time $t^\star \leq c_3/(\eta \mathbb{E}[\mathfrak{R}_n(\mathcal{F}_R, c_2)])$ such that*

$$\mathbb{E}[\mathcal{E}(g_{\alpha_{t^\star}}, \mathcal{F}_R)] \leq c_1 \mathbb{E}[\mathfrak{R}_n(\mathcal{F}_R, c_2)],$$

*where constants $c_1, c_2, c_3 > 0$ depend only on the boundedness constants $M, L$ and $R$.*

**In-sample linear prediction under $\ell_1$ constraints.** Let $Z \in \mathbb{R}^{n \times d}$ be a fixed-design matrix such that the $\ell_2$ norms of columns of $Z/\sqrt{n}$ are bounded by some constant $\kappa$. Assume a well-specified model, i.e., the existence of a vector $\alpha'$ such that the observations $y \in \mathbb{R}^n$ follow the distribution $y = Z\alpha' + \xi$, where $\xi$ is a vector with i.i.d. zero-mean $\sigma^2$-subGaussian components. We consider running mirror descent with the hyperbolic entropy mirror map [15] given by $\psi(\alpha) = \sum_{i=1}^d (\alpha_i \operatorname{arcsinh}(\alpha_i/\gamma) - \sqrt{\alpha_i^2 + \gamma^2})$ with any $0 < \gamma \leq (\|\alpha'\|_1 \wedge 1)/(3e^2 d)$ and $\eta \leq \frac{1}{24\kappa^2 \|\alpha'\|_1 \log(3\gamma^{-1})} \wedge \frac{\|\alpha'\|_1}{2\sigma^2}$. The theorem below yields minimax-optimal rates [31] up to the multiplicative factor $\log(3\gamma^{-1})$.

**Theorem 4.** *Consider the setup described above. There exists a data-dependent stopping time $t^\star \leq \sqrt{n}/(\eta \cdot 3\kappa\sigma\sqrt{\log d})$ such that with probability at least $1 - 2e^{-nc} - \frac{1}{8d^3}$, where $c$ is an absolute constant, we have*

$$\frac{1}{n} \|Z\alpha_{t^\star} - Z\alpha'\|_2^2 \leq 36 \cdot \frac{\kappa \|\alpha'\|_1 \sigma\sqrt{\log d}}{\sqrt{n}} \cdot \log(3\gamma^{-1}).$$

**Results on the whole optimization path.** Theorem 1 immediately implies that along its optimization path, continuous-time mirror descent satisfies excess risk guarantees that one would obtain for a series of ERM solutions over a corresponding set of *explicitly constrained* convex and bounded problems (cf. [39] for an exact setup) with varying radius $R$.

**Theorem 5.** *Fix any $\alpha_0 \in \mathbb{R}^m$, $R > 0$ and let $\mathcal{F}(\alpha_0, R) = \{g_\alpha : D_\psi(\alpha, \alpha_0) \leq R\}$. For any $\varepsilon > 0$, there exists a data-dependent stopping time $t^\star \leq 2R/\varepsilon$ such that for some $c_1, c_2 > 0$ depending only on the boundedness constants (cf. [39]), we have*

$$\mathbb{E}[\mathcal{E}(g_{\alpha_{t^\star}}, \mathcal{F}(\alpha_0, R))] \leq c_1 \mathbb{E}[\mathfrak{R}_n(\mathcal{F}(\alpha_0, R) - g_{\mathcal{F}(\alpha_0, R)}, c_2)] + \varepsilon.$$

## 5 Future Directions

Our work provides a simple and transparent framework for simultaneously analyzing statistical and computational properties of iterates traced by a family of mirror descent algorithms applied to the i.i.d. batch statistical learning setting. Among the research directions that would yield additional computational savings are extensions of our results to stochastic and accelerated frameworks, where connections between early stopping and localized complexity measures are yet to be established, even in the restricted setting of Euclidean gradient descent updates.

Beyond the computational savings, our main results reveal a curious property of mirror descent. For an unknown parameter of interest denoted by $\alpha'$, the statistical complexity of an appropriately stopped mirror descent iterate is given by the offset complexity of the class $\{g_\alpha - g_{\alpha'} : D_\psi(\alpha', \alpha) \leq D_\psi(\alpha', \alpha_0)\}$. Thus, $g_\alpha$ is *implicitly constrained* to lie in a *possibly non-convex* Bregman ball *centered at* the unknown $\alpha'$ with *unknown radius* $D_\psi(\alpha', \alpha_0)$. Therefore, in general, solutions traced by mirror descent iterates cannot be practically expressed as solutions of *explicitly constrained* optimization problems. Consequently, early-stopped mirror descent can potentially solve problems that cannot be tractably solved by the means of explicit regularization. This observation necessitates further investigation.

**Acknowledgments**

Tomas Vaškevičius is supported by the EPSRC and MRC through the OxWaSP CDT programme (EP/L016710/1). Varun Kanade and Patrick Rebeschini are supported in part by the Alan Turing Institute under the EPSRC grant EP/N510129/1.

## Broader Impact

This work does not present any foreseeable ethical or societal consequences.

## Footnotes

[1]For a full version of this paper see [39].

[2] In some cases, the results we obtain are not *exactly* comparable to the ones obtained in the related work, as some of our assumptions are considerably weaker, e.g. *non-realizable* setting, and our guarantees stronger, excess risk vs bounds in $\|\cdot\|_n^2$ or $\|\cdot\|_P^2$ norms defined in Section 1.1. However, in some applications we require boundedness which is not required in some of the prior work, and some of our results are stated only in expectation, rather than high probability. We note that we can also easily obtain high-probability results in some settings (e.g. heavy-tailed classes under the lower-isometry assumption) that are outside the scope of the related work in the early stopping literature. See the full version of this paper [39] for an extended discussion.

[3] If such a function $g_{\mathcal{F}}$ does not exist, we can redefine $g_{\mathcal{F}}$ to be any function in $\mathcal{F}$ such that $R(g_{\mathcal{F}}) \leq \inf_{g\in\mathcal{F}} R(g) + \delta$ for any arbitrarily small $\delta > 0$.

[4]The $\varepsilon$ term affects the resulting excess risk bounds only by an additive term equal to $\varepsilon$, which can be chosen to be arbitrarily small in our main results (cf. the full version of this paper [39]).

[5]When the parametrization is data dependent, such as in the setting of kernel methods, our main theorems also establish that the early-stopped mirror descent iterates satisfy the offset condition (cf. Definition 1). We analyze a concrete example and provide full details in Theorem 3.

[6]Since $\alpha'$ is independent of the data and since it corresponds to the best parameter in some class of interest, $R_n(\alpha') \approx R(\alpha')$ and hence it can be interpreted as the noise level of the problem.

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
