[Reviews · NeurIPS 2020]

Review 1

Summary and Contributions: This paper provides risk bounds when early stopping is employed for a family of mirror descent algorithms that satisfy a specific matrix structure wrt the data and predictor. With boundedness assumptions on the data, risk bounds are provided relative to stopping time (in the continuous case, or number of iterates in the discrete case) proportional to D/eps (where eps is desired accuracy and D is Bregman divergence of initial iterate to the point wrt which risk is evaluated); and Bregman divergences of iterates up to the stopping time are bounded. The continuous-time result is specifically applied to non-parametric (kernelized) regression and linear regression with norm constraints on design matrix columns. A bound on the excess risk in the general explicitly-regularized case (with the same structural and boundedness constraints) wrt stopping time and constraint parameters is provided, recovering in this setting established results for excess risk wrt offset Rademacher complexity.

Strengths: The characterization of iterate behavior relative to aptly-specified stopping times is useful, and the proof of these results is straightforward and clean. That this analysis obtains the known minimax optimal rate in the constrained regression setting of Theorem 4 is a nice connection to existing literature; and Theorem 5 meaningfully relates the preceding analyses to the explicitly-regularized setting.

Weaknesses: That the proofs of the main results (Lemma 1 and the specific instances presented in Theorem 3 and Theorem 4) consist of clean elementary calculations is largely due to the assumption that there exists a matrix Z encoding the predictor-data relationship such that the predictor can be conceived of as linear (i.e., that there is a matrix Z such that predictors {g_\alpha} parametrized by \alpha satisfy g_\alpha(x_i) = (Z\alpha)_i); this removes much difficulty in the analysis. While it is true that kernel methods and (by definition) linear prediction are important problems satisfy this assumption, it is nevertheless a very strong assumption that does not apply to many problems of interest, to say nothing of the drawbacks of kernelizing problems. The boundedness assumptions limit comparison to prior work, especially as those parameters will not grow particularly nicely with dimension. It is incomparable to the results of [Wei, Yang, Wainwright 19], which are parametrized by smoothness and strong convexity of the problem; however, the results there (and in the other works cited as points of comparison) are high-probability bounds rather than in expectation, and the conversion should have been made from the latter to the former to facilitate comparison. Similarly, though stopping times are best analyzed in continuous time, the results in Theorem 3 and Theorem 4 would be more meaningful converted to discrete time, as they are ultimately algorithmic.

Correctness: At this level of review, the results appear correct, though (as noted above) comparability to prior work is not as strong or direct as claimed.

Clarity: Yes, the paper is well written.

Relation to Prior Work: Prior work is sufficiently discussed, but the comparisons could be improved.

Reproducibility: Yes

Additional Feedback: [response to author feedback] Re: Theorems 3 and 4, I apologize for the oversight -- I missed the choices of step size wrt problem parameters. It would be useful to have them in the actual theorem statements, as they complete the respective theorems. My score remains unchanged because the more serious concern I identified, linearity vis-a-vis the matrix Z, appears to be a limitation inherent in this approach. I don't disagree that this covers many important settings; nevertheless it is a strong structural assumption that should be viewed (fairly presented) as a trade-off for the greater generality afforded elsewhere (eg, mirror vs gradient descent updates).


Review 2

Summary and Contributions: This paper develops a simple and novel framework for simultaneously analyzing statistical and computational properties of iterates traced by a family of mirror descent algorithms. The authors identify the link between mirror descent iterates and localized complexity measures. They also provide interesting results of bounds on the excess risk of the iterates of mirror descent using off set Rademacher complexities with applications. Overall, I feel this is a good work.

Strengths: This submission provides a novel perspective to study the complexity of early-stopped mirror descent.The idea is very well presented and authors provides rigorous theoretical analysis and make significant contribution. The authors also demonstrates the strength of their framework from different aspects with three applications.

Weaknesses: It would be nice if the authors could discuss the limitation of this method in more details, such as what assumptions that this framework uses, and which scenarios can apply this framework, and those scenarios cannot.

Correctness: Although I did not have a chance to check all details of the proof in the appendix, I feel this work is technically sound.

Clarity: Yes, this submission is overall well written and well organized. It might be better for authors to highlight all the assumptions used in their framework and compare the assumptions used in related works in the same paragraph.

Relation to Prior Work: Yes, the authors clearly discussed the related works in the section 2.1 Comparison with Related Work.

Reproducibility: Yes

Additional Feedback: Authors' rebuttal well addresses my concerns about the assumptions and limitations of this work. I keep my score unchanged because the author's response did not change my evaluation of the overall quality of this submission.


Review 3

Summary and Contributions: The paper studies early-stopped unconstrained mirror descent for linear models with squared loss. The authors identify a link between offset Rademacher complexity and potential-based convergence analysis of mirror descent which allows them to get excess risk guarantees. Their techniques allow them to simplify existing results on implicit regularization.

Strengths: Overall, the problem of understanding implicit bias of algorithms is super relevant to ML today and this paper makes an interesting step towards understanding the same. The connection between offset Rademacher complexity and mirror descent analysis (irrespective of the choice of the mirror map) is neat. The proof is clean and connects the two concepts essentially by definition. The fact that the framework captures various existing results in a unified manner is exciting.

Weaknesses: A weakness here is that the analysis seems heavily reliant on square loss (due to the use of offset complexity) and also on the linear assumption. It is unclear if it can generalize to convex functions. It would be good to discuss this further.

Correctness: I have checked the proofs of the main theorems carefully and they seem correct. However, I have not verified the proofs in Appendix D, so it is possible I might have missed something there.

Clarity: The paper is well-written and easy to follow. The proofs are precise and there is ample explanation. I like the notation table in the appendix. One suggestion would be to not refer to theorems before stating them (for example, Section 2.1). This makes it harder to really understand the differences.

Relation to Prior Work: The authors do a good job of comparing their results with prior work. They have been careful to mention the subtle differences between the settings in their work and that of prior work.

Reproducibility: Yes

Additional Feedback: Line 144 “R_n(\bar{\alpha}_t) can get arbitrarily close to R_n(\alpha’) for ant \alpha’” This is not exactly correct as you get a one sided bound only. Maybe word this differently? Line 314: Should it not be Z\alpha instead of \sqrt{Z} \alpha? Section D.2: [48] also uses a similar reparameterization to show implicit bias to optimal l1 solution. [Post Rebuttal Response] My score remains unchanged post the authors' response. The analysis being restricted to linear assumption and squared loss are valid limitations of the paper. Despite this, as my score reflects, I would like to see the paper accepted.


Review 4

Summary and Contributions: The paper studies the problem of early stopping for mirror descent algorithms when learning with a quadratic loss and provides statistical guarantees of the regularization implicitly obtained through early stopping.

Strengths: The problem authors address has very high impact: using early stopping for regularization has the benefit of reducing computation time for achieving better solutions to the problem. It is therefore a very popular technique among practitioners. Getting better understanding of the dynamics of early stopped regularizers for a wide class of problems (here covered using the mirror descent formalism) is therefore interesting. The work provides new results using the offset Rademacher complexity and linking it elegantly with mirror descent on the quadratic loss. Techniques used by authors to prove their main results are original and can be inspiring for future work.

Weaknesses: I did not see any information theoretical discussion on the optimality of the results stated. Are the bound in Theorem 1 optimal? can authors give examples in a simple scenario where lower t << D / epsilon violated the requirements? Also the main result arrives at the very end of the paper. This is because of the large number of concepts that needed to be reviewed in the paper. however at this point we feel like there is no more room for comparison with existing work. However several papers (cited by authors) address the early stopping problem in situations that are special cases of the mirror descent algorithm. It would have been good to see comparisons with existing results.

Correctness: The claims that I had time to dig into and verify looked correct to me. Question: [Section 1.1 Background] g_F is defined as a function in the class F that attains the infimum over a class of functions F of the risk R(g). Why does g_F exist? if it is an infimum it is not necessarily attained

Clarity: The paper is overall well written. It is a very dense paper with many definitions and notions introduced. Giving simple to follow examples throughout the paper could have helped making it more readable.

Relation to Prior Work: References that I was expecting to see were there and discussions establish the relation with the references. As noted in the "weaknesses" above, "several papers (cited by authors) address the early stopping problem in situations that are special cases of the mirror descent algorithm. It would have been good to see comparisons with existing results. "

Reproducibility: Yes

Additional Feedback:

[Author Response · NeurIPS 2020]

We thank the reviewers for valuable comments and questions. We address each reviewer individually below.

**Response to Reviewer #1.** Below we address the three weaknesses put forth by the Reviewer #1.

*1. On the linearity assumption.* The reviewer points out that the linearity assumption removes much difficulty in the
analysis. On the contrary, even for linear problems, the statistical/computational analysis of early-stopped iterative
algorithms (typically gradient descent) is rather involved and usually results in rather long proofs (see, e.g., [10, 37,
44, 47, 49, 50])). In our work, the analysis is so simple due to the novel connection that we develop between offset
Rademacher complexities and mirror descent algorithms. Given the fundamental nature of this connection, we hope
that our work will motivate follow-up research in this direction, extending our framework to non-linear algorithms and
also extending the framework of offset Rademacher complexities beyond the quadratic loss (see also response to R#3).

In addition, given the number of papers whose setups are special cases of ours (cf. Section 2.1 and Appendix D), we do
not see the linearity/kernel assumption as particularly limiting. While there is on-going implicit regularization research
in the context of neural networks (as we briefly review in Section 2.1), such works, in contrast to our work, do not focus
on excess risk guarantees. Finally, we emphasize that linear models are fundamental in statistics and machine learning,
and there is growing literature that uses them to also describe key features in non-linear neural networks. For instance,
see the double descent phenomenon and the Neural Tangent Kernel literature).

*2. On the boundedness assumption and comparison with [WYW, 19].* We remark that Theorems 1 and 2 can be
applied to also obtain high-probability bounds (cf. footnote 1 on page 3 and Appendix D.1). We opt to present bounds
in expectation as such results are the simplest to state and also require minimal assumptions on the data generating
mechanism, namely boundedness. Note that, in contrast to our results, [WYW] prove high-probability bounds under a
well-specified model assumption, which is not present in our work. In addition, the analysis of [WYW] does not apply
to derive results such as Theorem 4 (updates are not gradient descent) and Theorem 5 (statistical guarantees along the
whole optimization path) that can be easily proved within our framework. We refer to Appendix D.1 for an extended
discussion, where we also discuss why our results obtained in Theorem 3 are nevertheless similar to the ones in [WYW].

*3. On Theorems 3 and 4.* The reviewer's comments (both in summary and in weaknesses sections) on Theorems 3 and
4 is a misunderstanding since both theorems are **discrete** time results that follow via an application of Theorem 2. We
kindly ask the reviewer if they could have another look at Theorems 3 and 4 and the surrounding discussions.

**Response to Reviewer #2.** The primary limitations are reliance on the quadratic loss and linearity of the model.
We hope that both can be addressed in future work. Regarding the linearity assumption, please see our response
to the Reviewer #1. Regarding the quadratic loss, please see our response to the Reviewer #3. In the applications
that we present, the only assumption on the data-generating distribution is boundedness, which is necessary in the
distribution-free setting (i.e., in contrast to the related works [37, 47], we do not assume a true model generating the data).

**Response to Reviewer #3.** Regarding the linearity assumption, we point to our answer to the R#1 above. At this point,
it is indeed unclear under what conditions our arguments generalize to other loss functions. We believe that there would
not be significant difficulties in extending our arguments to smooth and strongly convex loss functions considered in
[47]. At the same time, we believe that the fundamental connection that we have observed between mirror descent and
offset Rademacher complexities *can offer a dual view on localization via offset Rademacher processes*, facilitating
future research in this direction. Regarding the reviewer's suggestions, we agree with all of them.

**Response to Reviewer #4.** Regarding the information-theoretic discussions, localized complexity measures are known
to yield minimax-optimal rates for various problems. In addition to the early works on localized complexity measures
[9,23], Corollary 12 in [25] shows that offset Rademacher complexities capture correct rates for non-parametric regres-
sion. In Section 3.3 in [47], minimax optimality of localized complexities is established for regular kernels. We will add
these pointers. The function $g_{\mathcal{F}}$ such that $R(g_{\mathcal{F}}) = \inf_{g \in \mathcal{G}} R(g)$ is defined as a notational convenience (f.note 2 on p. 3).
Finally, we address the reviewer's questions regarding the stopping time $t^\star$ and the
request for a simple example demonstrating the key concepts that appear in our paper.
Intuitively $t \ll t^\star$ (where $t^\star$ is the prescribed stopping time) results in a sub-optimal
estimator for the same reason that poorly tuned explicitly regularized estimators are
sub-optimal (i.e., the number of iterations $t$ play a similar role to the regularization
parameter $\lambda$ in explicitly penalized problems). Consider a data model $x_i \sim N(0, I_d)$ and

(a)   (b)

$y_i | x_i \sim \langle \alpha', x_i \rangle + N(0, \sigma^2)$ and assume that $\alpha'$ is a sparse vector. Due to the sparsity of
$\alpha'$, lasso is superior to ridge regression (cf. Fig. (a)); similarly, mirror descent with the
hyperbolic entropy potential is superior to gradient descent (cf. Fig. (b)). Also note the
similarities between Fig. (a) and (b). Finally, Fig. (c) and (d) graphically demonstrate the
main idea behind our proof techniques: up to the vertical dotted line that denotes $t^\star$ (cf.
line 277), the Bregman divergence $D_\psi(\alpha', \alpha_t)$ is non-increasing (green lines). **[Zoom**

(c)   (d)

**in on the figure for details.]**

[Meta-Review · NeurIPS 2020]

The paper provides a clean and simple analysis of early-stopped mirror descent via a connection to offset Rademacher complexity. As noted by several reviewers, the downside is that linearity and square loss are essential for the proof to go through, but given the interest in the topic and approachability of the proofs, the paper will be of value to the broader NeurIPS community.